# Geographical Distance, Socioeconomic Deprivation, and Educational Level Shape Access to Voluntary Termination of Pregnancy in a Southern Region of Italy [note 1]

**DOI:** 10.3390/healthcare13172160

**Published:** 2025-08-29

**Authors:** Nicola Bartolomeo, Letizia Lorusso, Maria Carella, Roberta Pace, Paolo Trerotoli

**Affiliations:** 1Interdisciplinary Department of Medicine, University of Bari Aldo Moro, Piazza Giulio Cesare 11, 70124 Bari, Italy; paolo.trerotoli@uniba.it; 2School of Medical Statistics and Biometry, Interdisciplinary Department of Medicine, University of Bari Aldo Moro, Piazza Giulio Cesare 11, 70124 Bari, Italy; letizia.lorusso@uniba.it; 3Department of Political Sciences, University of Bari Aldo Moro, Piazza Cesare Battisti 1, 70121 Bari, Italy; maria.carella1@uniba.it (M.C.); roberta.pace@uniba.it (R.P.)

**Keywords:** voluntary termination of pregnancy (VTP), socioeconomic deprivation, healthcare access, abortion, travel time, geographical distance, catchment area

## Abstract

**Background**: In Italy, voluntary termination of pregnancy (VTP) is a legally protected healthcare service. However, in Apulia, a southern region, access remains uneven due to ongoing healthcare rationalization, which has reduced service availability, particularly in decentralized areas. Conscientious objection among providers may also contribute, although the number of VTPs per provider has decreased over time. This study examines whether women access VTP services outside their healthcare catchment area (CA) and how socioeconomic deprivation and individual factors may influence mobility. **Methods**: We applied a ranking method, based on spatial and temporal distance between hospitals and municipalities to define the catchment area (CA) around hospitals of the Apulia region that offers VTP service. A Poisson multivariable clustered model was applied to evaluate the association among demographic and socioeconomic factors and the choice of the VTP service. **Results**: The analysis revealed that 54.7% of VTPs were performed outside the women’s catchment area. This mobility was significantly more frequent among women from medium and low socioeconomically deprived areas compared to very low deprived areas (RR = 1.20; 95%CI: [1.02–1.42]) and (RR = 1.28; 95%CI: [1.03–1.57]). Higher education level (RR = 1.09; 95% CI: [1.04–1.14]) and employment (RR = 1.09; 95%CI: [1.03–1.14]) were also associated with higher rates of undergoing a VTP outside of CA, with variations observed across local health authorities. **Conclusions**: These findings have shown the influence of socioeconomic conditions and educational level on women’s access to VTP services, suggesting that structural inequalities continue to shape healthcare choices and mobility.

## 1. Introduction

In Italy, voluntary termination of pregnancy (VTP) is governed by Law 194/78, which establishes a structured framework aimed at balancing women’s reproductive rights with medical and ethical safeguards. The first step is an initial consultation with a general practitioner or gynecologist and/or in a family counseling center, which provides comprehensive information on pregnancy options, procedural details, and health risks, offering psychological support if needed. Next, the woman achieves a certification of the pregnancy state and has the right to perform a voluntary interruption. The certification is followed by a period typically lasting 7 days, during which the woman can reconsider her decision. This time interval could be avoided in case of risks to the woman’s health, pregnancy at risk of fetal death, violence, or pregnancy resulting from crimes [1].

The woman is then guided in the choice of a facility, typically a Gynecology and Obstetrics Operating Unit within an accredited public or private hospital. The certification must be submitted to the healthcare facility that carries out the procedure.

Clinical assessments confirm the gestational age and evaluate the woman’s physical and psychological conditions, after which she may choose between public and private accredited facilities. Once all steps have been completed, the VTP procedure is performed in an authorized healthcare facility. Depending on the circumstances and the stage of the pregnancy, the termination can be either surgical or medical (Figure 1). At the end stage, the woman is monitored during a follow-up period to check for any complications and ensure her recovery. The care pathway is structured to ensure equitable access and comprehensive support for women pursuing VTP, through the adequate territorial distribution of family counseling centers and authorized hospitals [1].

This process ensures that the woman is well-informed and supported throughout the decision-making and medical steps leading to the voluntary termination of pregnancy.

Between 2019 and 2020, Italy experienced the most significant decrease in the number of VTPs (total rate of VTPs per 1000 women resident aged 15–49 from 218 to 191), a marked decline likely attributable to the COVID-19 pandemic. In the following period, a steady but gradual increase was observed between 2020 and 2021 (from 191 to 194) and again from 2021 to 2022 (from 194 to 206) in rates of VTPs per 1000 women. This pattern has multiple explanations. When considering total VTP rates per 1000 women resident, a general decrease was recorded across all major Italian macro-areas. In addition, both the 2021 and 2022 updated reports confirm a continued stabilization and overall decline in abortion rates across Italy, despite minor regional differences and the transient effects observed during the pandemic period [2,3,4].

Ultimately, both the official reports updated to 2021 and 2022 and the synthesis work published by de Fazio et al. [3] confirm a consistent decline in Italy over the years (from 1983 to 2019) [2,3] in the total VTP rate per 1000 women, with some regional exceptions [4]. For instance, in 2022, the highest standardized rates of VTPs among resident women aged 15–49 (per 1000 women) were recorded in Liguria (8.43), followed by Piedmont (7.07), and Emilia-Romagna (6.88). Apulia ranked fourth, with a rate of 6.86—just 0.02 points below Emilia-Romagna [2,4].

Essential organizational factors include guaranteeing individuals the right to choose any public or accredited private service across the country, ensuring an adequate number of Gynecology Operating Units in proportion to the population’s needs, and maintaining facility accessibility. However, the downsizing of healthcare structures as a result of cost-containment policies has reduced the number of available services, highlighting the need to re-evaluate users’ actual access to care. The number of objectors (health care professionals who declare they do not practice VTP) may play a role in choice and accessibility. The percentage of professional conscientious objectors in Apulia in 2021 was 80.6% of gynecologists, 41.4% of anesthetists, and 67.5% of nurses and midwives involved in the VTP procedure [2], with minimal fluctuations over the years. 

Existing VTP services can still serve as reference points for providing services to a specific population residing within a defined geographic area administered by a local health authority. It is crucial to highlight the right to free choice of the unit, implying a degree of users’ mobility towards facilities that may be located in different areas from their residence. The reasons for such mobility include factors such as the waiting list length, service quality, accessibility, privacy, and personal considerations [5].

The decision-making process of women requesting VTP is shaped by a combination of individual factors, such as age, educational attainment, socioeconomic status, and religious affiliation, and contextual factors related to the availability and accessibility of services in their area of residence [6,7]. This approach, based on the Andersen Behavioral Model of healthcare utilization [8], shifts the focus from family units to individuals, highlighting the influence of population characteristics and health behavior on the use of maternal health services. Women facing the emotional challenge of pregnancy termination are influenced by multiple factors, including cultural and educational background, religious beliefs, and the availability and quality of healthcare facilities and services in their area of residence [7]. Many women’s perceptions are shaped by concerns regarding the quality of care available in provincial healthcare facilities compared to those in the regional capital. Additionally, choices about where to seek care are significantly influenced by the perceived expertise of medical staff and the level of privacy that different facilities are able to guarantee.

A main factor to consider is women’s level of education and employment status, as these two conditions provide greater autonomy and access to financial resources. This increased independence enables women to seek care outside their designated catchment area and to choose facilities that offer higher standards in both medical treatment and the personal management of pregnancy termination. Moreover, health literacy and informed reproductive decision-making are closely linked to greater awareness of available healthcare options, both public and private, allowing women to make more conscious and informed choices [9,10,11]. Economic status plays a pivotal role in shaping healthcare decisions. When transportation is affordable, the physical distance from home becomes less of a barrier, and factors such as the quality of care and associated costs take on greater importance in determining facility choice.

Other studies have underscored the link between population socioeconomic status and mobility patterns to healthcare access points [10]. 

Many studies have highlighted the importance of evaluating service catchment areas by taking into account not only the spatial relationship between users and service locations but also the actual travel time required to access these services. Contemporary research emphasizes that analyzing catchment areas solely based on straight line distance can be misleading, as it fails to reflect real-world accessibility influenced by transport networks, road conditions, and available modes of travel [12]. Research from various contexts has shown that longer travel distances can reduce access to care and contribute to unmet health needs. For example, Thompson et al. (2021) [5] found that greater distance to abortion services in the US was linked to lower abortion rates, while studies in Sweden and Rwanda highlighted barriers to reproductive healthcare and the potential role of telemedicine in improving access [13,14].

In the Italian context, however, there is a notable scarcity of region-specific analyses. One exception is the study by Facciolà et al. (2018) [15], which examined women’s decisions to undergo voluntary termination of pregnancy (VTP) in Sicily, focusing on risk factors in the absence of fetal abnormalities. Nevertheless, few studies have assessed how socioeconomic and geographic inequalities interact to influence access to VTP services at the regional level, and how these patterns vary within the decentralized Italian healthcare system.

The CA is defined as interconnected zones where the population travels along suitable transport routes to a central point and is calculated considering factors such as the distance decay within the catchment area, variable catchment area sizes, and the availability of transportation modes that influence access to healthcare services [16,17]. For healthcare structure accessibility, catchment areas refer to the regions from which patients travel to visit that facility [16]. Concurrently, since users are free to choose their preferred service, it is generally expected that they will opt for the facility offering the greatest accessibility in terms of distance and travel time. Nevertheless, individual empowerment can prompt different behaviors, with some users choosing to travel to facilities that are less accessible or located farther away, driven by personal preferences or specific needs that outweigh considerations of proximity.

This paper aims to examine the factors influencing women’s choices regarding healthcare access. Specifically, the study investigates whether users tend to select hospitals with VTP service within their geographical area of residence, and whether the socioeconomic deprivation of their residential area, along with individual sociodemographic characteristics, are associated with mobility towards healthcare access points.

This article is a substantially expanded and revised version of our earlier work presented as an abstract at the 52nd Scientific Meeting of the Italian Statistical Society, published in the conference proceedings [18]. All analyses, results, and the discussion presented here are original and were not included in the previous version.

## 2. Materials and Methods

### 2.1. Data

To explore the link between women’s mobility for accessing abortion services and the level of socioeconomic deprivation in the Apulia region, a retrospective observational study was conducted. The analysis focuses on all VTPs carried out between 1 January and 31 December 2022, across both public and private hospitals with VTP service in the region as collected in the Apulia’s Regional Health Information System according to the National Statistical Plan [19] and managed by National Institute of Statistic (ISTAT). Data were anonymized prior to analysis with the standards set by the National Institute of Statistics (ISTAT) and processing was conducted in full compliance with the provisions of Article 24 of the EU General Data Protection Regulation (Regulation (EU) 2016/679). The form to collect data on a VTP consists of two sections: the first contains general details about the woman and the pregnancy, while the second includes information about the termination of the pregnancy. We considered all variables that could potentially determine the primary endpoint, namely the choice of undergoing VTP within or outside one’s own catchment area. From the first section, we included the age at pregnancy, municipality of residence, citizenship, marital status, education level, parity, gestational age, weeks of amenorrhea, and the presence of fetal malformations. From the second section, we considered the service that issued the certification (family counseling service or other), the type of facility (public or private) where the VTP was performed, the time elapsed between certification and the procedure date, the urgency, and the type of VTP (pharmacological, surgical, or other).

To obtain the most comprehensive information, some variables were dichotomized into risk groups that could better reflect the observed relationships. Age at pregnancy was categorized into two groups based on the cutoffs used in the Ministry of Health report [2]; then, we opted to dichotomize age using 30 years: women under 30 years of age and those aged 30 or older. Marital status was classified as either “married or in a non-marital union” or “not in a couple,” which included single, widowed, separated, or divorced individuals. Parity was used to distinguish between women with children and those without. Employment status was divided into two categories: “employed” and “unemployed,” with the latter group including unemployed individuals, those seeking their first job, housewives, and students. Educational level was categorized into two main groups: women with at least a diploma or higher qualification (ranging from a university degree to other higher education qualifications) and those with lower education. Weeks of amenorrhea were grouped into “<9 weeks” and “≥9 weeks.” The interval between certification and procedure completion was divided into “<15 days” and “≥15 days,” while the gestational age was categorized as “<90 days” and “≥90 days.”

VTP cases with missing data in any analyzed covariates were excluded, although the missingness mechanism could not be definitively determined for each individual variable. This assessment involved comparing results from a complete-case dataset and an imputed dataset—using Fully Conditional Specification (FCS) for categorical variables—and conducting sensitivity analyses that affirmed the robustness of results across both datasets (see Appendix A). Given a large sample size (only 3% of incomplete cases), we therefore based final model estimation on the complete-case dataset to avoid potential imputation-induced noise and preserve estimate precision [20]. Additionally, a missingness pattern matrix is provided in the Appendix A to enhance transparency regarding the extent and structure of missing data.

### 2.2. Determination of the Catchment Area

Women’s mobility was assessed using a binary variable that distinguished between cases where the voluntary termination of pregnancy (VTP) was performed at the designated reference facility for the woman’s municipality of residence and cases where care was sought at a hospital outside the relevant catchment area (CA). Across the region, VTP services are offered by 16 hospitals.

To delineate the CA for each hospital, both spatial and temporal distances were calculated between the urban centers of all 257 municipalities in the region and each of the 16 VTP hospital centers. Spatial distances (in kilometers) and travel times (in minutes) were determined using the Google Maps platform, which identified the centroid of each urban area and measured distances to the exact address of each hospital.

For every municipality–hospital pair (i,j), we obtained a spatial $\left(d_{spatial}\left(i,j\right)\right)\;$ and temporal distance $\;\left(d_{temporal}\left(i,j\right)\right).$ For each municipality, the resulting set of 16 distances (one per facility) was ranked in ascending order, separately for spatial and temporal values. This produces a spatial $\;r_{spatial}\left(i,j\right)\;$ and temporal $\;r_{temporal}\left(i,j\right)$ rank. The total score for each pair was calculated as the sum of the two ranks (1):(1)$$S\left(i,j\right)=r_{spatial}\left(i,j\right)+\;r_{temporal}\left(i,j\right)\;$$

Each municipality was then assigned to the hospital with the lowest combined rank $S\left(i,j\right)$, which was considered its reference structure. The CA of each facility was thus defined as the set of municipalities for which that hospital represented the minimum-rank option in terms of the combined spatial and temporal proximity.

To assess the robustness of the catchment area (CA) assignment with respect to potential variability in travel time estimates (e.g., due to traffic conditions, mode of transport, or time of query), we conducted a Monte Carlo sensitivity analysis.

Firstly, a random sample of 50 municipality–hospital pairs was extracted from the dataset, and travel times were recalculated using Google Maps at a different time point. For each pair, the relative difference in travel time was computed, and the mean of these relative differences was −0.00076, and the standard deviation was 0.029. Secondly, assuming that these differences approximate the variability in travel time estimation, we simulated 1000 perturbed datasets by applying a normally distributed random error term (N(−0.00076, 0.029)) to the original travel times. For each simulation, we redefined the CA for each municipality based on the modified times, following the same procedure used for the original assignment. For each municipality, we recorded the percentage of simulations in which its assigned CA changed. The results showed that in 75% of municipalities, the reference facility remained unchanged across all simulations. At the 90th percentile, the 3% of simulations resulted in a change of catchment area; at the 95th percentile, fewer than 16% of simulations resulted in a change of CA, and even at the 99th percentile, this proportion remained below 43%. These findings support the robustness of the CA definitions adopted in the main analysis.

Figure 2 illustrates the geographic distribution of the 16 VTP hospitals across Apulia and the corresponding catchment area (CA) assigned to each, with municipalities color-coded by reference hospital.

Each dot represents one of the 16 hospitals providing VTP services. Municipalities were color-shaded according to the facility assigned as their catchment area based on the minimum combined spatial and temporal distance from the urban centroid. This map visualizes the territorial allocation of services and highlights the regional organization of VTP accessibility.

### 2.3. Level of Socioeconomic Deprivation

Socioeconomic deprivation at the geographic area level was assessed using the deprivation index (DI) by Caranci revised by Rosano et al. in 2011 [21] that introduces methodological improvements, including a redefinition of indicators on low education and single-parent households to better reflect their analytical purposes. The deprivation index (DI) captures the relative socioeconomic disadvantage of the population within a given geographic area, drawing on census data. It is constructed using a set of indicators selected to reflect different dimensions of social and economic hardship, including low educational attainment (proportion of the population with an education level equal to elementary school, literate or illiterate, among the population aged six and over), job shortages (proportion of individuals who are unemployed or seeking their first job among the total workforce), poor housing (proportion of homes occupied by renters among all homes occupied by resident persons; ratio of the total population to the surface area of dwellings occupied by resident persons), and family conditions (single father or mother with children, in single nuclear families, with and without isolated members among the total families). 

For this analysis, all women residing within the same area were assigned an identical DI value. The DI is a continuous variable, and in this study it was classified into quartiles, based on evidence that deprivation levels remain relatively stable over time. Caranci et al. (2010) [22] analyzed deprivation trends over a decade and found a strong correlation between the DI of 1991 and that of 2001. Building on this evidence, we used the 2011 DI as a proxy for 2022 data and categorized it into quartiles to account for potential variations, as the likelihood of a municipality shifting to a different quartile over ten years is low.

### 2.4. Statistical Analysis

Continuous variables were summarized as means with standard deviations (SD) for those following a normal distribution, or as medians with interquartile ranges (IQR) when the distribution was non-normal. Normality was evaluated using the Kolmogorov test. Categorical variables were compared between women residing inside and outside the catchment area (CA) using the chi-square test. 

A multivariable Poisson regression model was used to explore the association of DI and the proportion of VTPs performed outside the area of residence (rate of VTPs outside the CA), independently of individual and VTP-related characteristics. Since the data were structured as aggregated counts for each combination of age group, educational level, employment status, and type of VTP within each municipality, the model accounted for intra-municipality correlation by applying a population-averaged approach with clustering at the municipality level. This adjustment ensures valid inference by correcting standard errors for potential within-cluster dependence.

To verify the adequacy of the Poisson model, we assessed overdispersion by comparing the deviance and Pearson chi-square statistics to their respective degrees of freedom. Both ratios were below 1 (deviance/DF = 0.653; Pearson χ^2^/DF = 0.560), indicating the absence of overdispersion.

Additional contextual covariates were included to reflect potential influences on women’s decisions to seek abortion services outside their CA. Specifically, we considered the local health authority (LHA) to which each municipality belongs and the number of reference facilities available in each municipality. Variable selection for the multivariable Poisson regression model was performed using a stepwise procedure with entry and removal thresholds based on *p*-values (<0.15). The final model also had the lowest Quasi-likelihood under the Independence Model Criterion corrected (QICu), indicating optimal fit among the candidate models. To assess potential multicollinearity, we created dummy variables for all categorical covariates and calculated Variance Inflation Factors (VIFs) using a linear regression approach. All VIFs were well below 10, confirming the absence of problematic collinearity. Results from the multivariable model are presented as rate ratios (RRs) with 95% confidence intervals, comparing the rate of VTPs performed outside the CA across various categories of each variable. Multiple comparisons were adjusted by Tukey. All statistical significance tests had two-tailed alternative hypotheses, and *p*-value < 0.05 was considered statistically significant. SAS/STAT® version 9.4 was used to carry out statistical analyses.

## 3. Results

In 2022, in Apulia 5316 women chose a VTP. We selected the 4997 (93.6%) women who resided within Apulia’s municipalities, excluding non-residents who underwent abortion in a hospital located in Apulia. A total of 4858 women were included in the complete-case analysis. The previous comparison between the complete-case dataset and imputed dataset indicated that the decision criterion based on *p*-values remains invariant across both datasets in the explorative analysis (Appendix A), and that the resulting estimates and conclusions are congruent Appendix A. We observed no meaningful differences in the exp(β) estimates between the complete-case and imputed datasets. In the Poisson regression for the number of structures within the CA, confidence intervals (CI) and standard errors (SE) narrowed slightly (e.g., from [0.57–0.74] to [0.78–0.9] and from 0.07 to 0.03, respectively), while exp(β) increased from 0.45 to 0.84. Although some variation in point estimates was noted, the directionality of all associations remained unchanged, and increases in estimates across the imputed dataset did not exceed 10%, confirming the robustness of the results in both cases. Among the 4858 women in the complete-case dataset, the median of age of the women that sought a VTP was 31 years [IQR: 24–36], and the median gestational age at the time of seeking a VTP was 7 weeks [IQR: 6–9]. 

The VTP procedure could be performed in 16 hospitals along all the 6 health districts (managed by LHA), 13 hospitals are public that treated 4093 (84%) women, while 3 private treated 765 (16%) women.

A total of 2657 women (54.7%) sought services outside their CA. To describe the profile of women that underwent a VTP and to evaluate differences between those treated inside CA compared to those outside, an exploratory univariate analysis of the sociodemographic and clinical variables was performed. This analysis showed significant differences between women treated outside CA compared to those treated inside CA for instruction degree (58.6% vs. 51.5%, *p* < 0.0001) and proportion of employed women (43% vs. 30.8%, *p* < 0.0001). Many individuals in both groups were Italian, but the proportion of non-Italian citizens was slightly higher inside the CA compared to outside the CA (11.0% vs. 8.3%, *p* = 0.0016, respectively). No significant differences were observed by age class (*p* = 0.9711) and marital status (*p* = 0.2456). In terms of the socioeconomic context, the majority of VTPs (59%) involved women residing in municipalities with a high deprivation index. However, a higher proportion of women undergoing VTP outside the catchment area lived in municipalities with a medium deprivation index (26.2% vs. 14.4%), whereas those using facilities within the catchment area more frequently came from municipalities with very low (7.5% vs. 2.2%) or low (19.5% vs. 12.2%) deprivation indices, and the differences were statistically significant (Table 1).

Women who underwent VTP outside the catchment area were more likely to access private healthcare facilities (18.5% vs. 12.3%, *p* < 0.0001), and the results showed that those who choose hospitals outside CA have a lower percentage of cases with access to the procedure within 15 days of certification (8.9% vs. 16%, *p* < 0.0001). Differences also emerged in the certification process, with a higher proportion of women undergoing VTP outside the catchment area obtaining certification from family counseling center (75.8% vs. 69.9%, *p* < 0.0001), whereas those within the catchment area were more likely to receive certification from a general practitioner or other facilities. The type of VTP also varied significantly, with pharmacological termination being more frequent among those treated outside the catchment area (61.4% vs. 43%, *p* < 0.0001), while surgical procedures were more common within the catchment area. Urgency was reported more frequently among women undergoing VTP outside the catchment area (48.8% vs. 40.2%, *p* < 0.0001). Moreover, while many procedures occurred before 90 days of gestation in both groups, this proportion was slightly lower among women treated outside the catchment area (92.2% vs. 95.1%, *p* < 0.0001). Finally, among pregnancies with fetal malformations, a higher proportion were carried out outside the CA (5.7%) compared to within the CA (3.4%), with a *p*-value of 0.0001 (Table 2).

To evaluate the independent association of sociodemographic and clinical characteristics of the women, as well as organizational characteristics of the LHA (number of structures within the CA, availability of pharmacological procedures, type of certification, and time from certification to procedure), with the choice to be treated outside the CA (the outcome variable), we used a multivariate Poisson model. The final model included the LHA, educational attainment, employment status, gestational age, marital status, and the deprivation index class.

In the LHA category, the highest rate of mobility outside the catchment area was observed in BA with a rate of 93.63 (95% CI: 82.25–106.59), followed by BR with a rate of 71.37 (95% CI: 61.91–82.28). Conversely, the lowest rate was recorded in LE, with a rate of 16.04 (95% CI: 10.59–24.31). When considering educational attainment, individuals with a diploma or degree had a rate of 54.13 (95% CI: 48.73–60.13), compared to 49.54 (95% CI: 44.32–55.38) for those with other forms of education. Similarly, employment status showed employed individuals having a higher rate of 53.96 (95% CI: 48.27–60.31) compared to 49.70 (95% CI: 44.67–55.29) for others. In terms of gestational age, there was a noticeable difference between those with gestational age < 90 days, showing a rate of 46.97 (95% CI: 42.25–52.22), and those with ≥90 days, who had a rate of 57.09 (95% CI: 49.29–66.12). Marital status also played a role, with married or non-marital union individuals having a rate of 50.20 (95% CI: 45.06–55.92) compared to 53.42 (95% CI: 47.94–59.53) for others. The deprivation index indicated variability, with low-deprivation index groups showing the highest rate (Figure 3).

In the regression models, we set the following reference categories: “High DI” for the deprivation index; single (or, more generally, unmarried) women for marital status; “Taranto (TA)” for the local health authority; VTPs sought ≥ 90 days after presentation, for time until voluntary termination of pregnancy (VTP); unemployment, student status, etc., for employment status; and the lowest educational qualification as the baseline for education. Women with low and medium deprivation indices showed significantly higher RRs when compared to those with a very low-deprivation index, with RRs of 1.20 (95% CI: 1.02–1.42) and 1.28 (95% CI: 1.03–1.57), respectively. However, the high vs. very low-deprivation index comparison did not show a significant association, with an RR of 1.07 (95% CI: 0.70–1.63). Other significant findings included a lower RR for married or non-marital union individuals compared to singles, with an RR of 0.94 (95% CI: 0.90–0.98), and for gestational age < 90 days vs. 90+ days, with an RR of 0.82 (95% CI: 0.71–0.95). Employment status comparisons revealed a significant increase in RR for employed individuals, with an RR of 1.09 (95% CI: 1.03–1.14), and educational attainment showed higher RRs for those with a diploma or degree, with an RR of 1.09 (95% CI: 1.04–1.14). The number of hospitals with VTP centers within the CA also seems to be associated with a decrease in RR for each additional structure, RR = 0.65 (95% CI: 0.51–0.84) (Figure 4).

## 4. Discussion

This study examines women’s tendencies to choose local healthcare facilities for the VTP and investigates how the socioeconomic deprivation of residential areas and individual demographic factors correlates with the mobility towards VTP access points among the local health authority in Apulia. Previous studies explored similar issues concerning access to health care services and patient mobility, particularly in relation to reproductive health services [23,24,25] or to maternal health services in different countries [24]. However, our study is one of the first to specifically address these dynamics within the Italian VTP healthcare context, in the Apulia region, where high rates of conscientious objection and regional disparities in service availability may influence patient decisions [26,27]. The work of Pecoraro et al. [26] outlined the distribution of intra and interregional services in Italy to capture the level of equity across the country, showing an unequal distribution of high-quality resources between the north and south of Italy. Different works underline the differences in access to VTPs’ healthcare services in different countries, showing how these differences affect the possibility of obtaining a VTP in due time [28,29] or the regional availability and accessibility [23]. 

Understanding the accessibility of healthcare services is crucial for assessing the equity in service distribution and utilization. The concept of a catchment area (CA) is widely used in healthcare research to define the geographical area from which a facility draws its patients [16]. Traditional methods for defining CA often rely on administrative boundaries, predefined radius distances, or patient-flow data derived from hospital admissions. In general, spatial healthcare access measurement has primarily focused on two main methods: the Two-Step Floating Catchment Area (2SFCA) method and the Kernel Density (KD) method, which have both undergone methodological development and been applied in empirical studies [17]. While these approaches provide a structured framework, they do not always capture real-world accessibility, as they fail to consider the real travel times and distances that patients experience. In our study, we defined the catchment area by calculating the real spatial and temporal distance traveled by women who underwent VTP, based on a spatial and temporal matrix of distances built on the geographical and temporal distances allocated from the centroid of each municipality and the address of the facilities in the Apulia region. This method, also applied in other works [25,30], provides a more dynamic and individualized representation of accessibility, reflecting the actual burden of travel more accurately than conventional models. Similar approaches, which define healthcare catchment areas based on travel times rather than administrative boundaries, have been applied in prior studies assessing geographic accessibility and service distribution [31,32]. Challenges in obtaining precise individual locations (due to factors such as anonymization, data accuracy [33], or geocoding limitations [34]) often necessitate using proxies. Our study employed municipality centroids as a proxy for women’s location. We acknowledge this as a limitation, particularly in larger or rural municipalities where intra-municipal variability in accessibility may be more pronounced. However, this simplification represents a common and practical trade-off, as such proxies (e.g., postal code centroids) have been shown to approximate true locations reasonably well in many population-based studies [35], balancing spatial precision with data availability and reliability [32,33]. To reinforce the robustness of the catchment area assignment despite this limitation, we conducted a Monte Carlo sensitivity analysis simulating variability in travel time estimates. The results showed high stability in CA attribution across simulations, supporting the reliability of our method even in the presence of small fluctuations in estimated travel times.

Furthermore, the choice of a binary outcome based on CA assignment, rather than using continuous travel distance or time, aligns with the study’s primary aim: to inform health service planning by identifying gaps in territorial coverage and ensuring that organizational decisions are guided by clearly defined geographical responsibilities. In this perspective, the binary classification of being inside or outside a CA is not only analytically sound, but also policy relevant.

In Apulia, the prevalent choice among women is a public hospital. Specifically, 82% of women choose a public hospital outside their catchment area (CA), compared to 87% who opt for a public hospital within their CA. Women that reside in the area managed by local health authorities of Bari, Brindisi, and Taranto choose to go outside their reference structure compared to those who remain within it. Conversely, those who reside in the area managed by the local health authorities of Lecce, Foggia, and Barletta–Andria–Trani have shown an opposite trend. This difference among LHAs may be partly explained by the distribution and density of available VTP facilities. In urban areas such as Bari, where multiple facilities are located within close proximity, women may access services outside their assigned CA while remaining within a short geographical radius. These patterns reflect not a failure of the CA assignment model but rather illustrate healthcare mobility within areas of high service density, precisely the type of behavior that our approach is designed to capture. Thus, higher mobility rates probably reflect greater opportunity to choose within a densely served area. 

Another possibility that could explain the choice is the availability of pharmacological abortion in each LHA. On 2020, in Italy, AIFA Decision No. 865 [2,36] extended access to medical abortion up to 63 days of amenorrhea (9 weeks of pregnancy). The Report of the Ministry of Health on the Implementation of the Law on Social Protection of Maternity and Voluntary Termination of Pregnancy shows a progressive increase in distribution from 2015 to 2018, stabilizing until 2020, followed by a significant rise through 2022; each increase coincides with corresponding AIFA resolutions. In addition, Ulipristal acetate sales rose by 27.7% from 2021 and 66.8% from 2020, following the 2020 AIFA resolution removing the prescription requirement for minors. Levonorgestrel (Norlevo) sales increased by 6.7% from 2021 and 4.8% from 2020. Due to untraceable sales data, usage by age group and repeated use cannot be assessed [2].

However, not all hospitals with VTP services have promptly adapted to this change; as a result, access to medical abortion services still varies across different areas of the region. This uneven availability likely influences women’s decisions to travel to hospitals that offer pharmacological abortion.

Women often prefer the medical approach due to its perceived naturalness, the possibility of undergoing the procedure in a private setting, and the desire to avoid surgery or anaesthesia [37]. A systematic review and several qualitative studies have shown that women’s choices are shaped not only by clinical information but also by emotional factors, previous experiences, and the perceived level of control over the process [37,38,39]. The concept of spatial equity and access to healthcare services has been addressed in several papers [23,30,40,41].

The role of socioeconomic deprivation in shaping access to women’s healthcare services has been widely documented [42,43,44]. Our findings indicate that women residing in municipalities with medium and low-deprivation index were more likely to seek voluntary termination of pregnancy (VTP) services outside their catchment area (CA). This pattern may reflect a multifaceted dynamic. While previous studies have shown that deprivation is associated with disparities in access to reproductive health services and contraceptive provision [43,44], the specific mechanisms influencing mobility for abortion care remain underexplored. One possible explanation is that women in medium and low-deprivation areas may have fewer nearby facilities offering VTP services or may perceive local services as insufficient in terms of quality or confidentiality. Additionally, the reputation and perceived reliability of larger, centrally located, or private hospitals may act as a pull factor, encouraging women to bypass local services, a phenomenon observed in other areas of healthcare utilization [45,46].

However, other hypotheses, such as limited awareness of available services, reduced health literacy, or differing perceptions of institutional trust, have not been directly examined in the context of abortion care and should be investigated in future research. Understanding whether informational, cultural, or organizational barriers contribute to these mobility patterns could inform targeted interventions aimed at reducing inequalities in access to VTP services.

Regarding educational level, our study found that women with a diploma or degree were more likely to seek VTP services outside their CA. This finding is consistent with the previous research, suggesting that higher education levels may lead to greater awareness of healthcare options and the ability to navigate healthcare systems more effectively [8,34,35]. Similarly, employment status was significantly associated with mobility. Here, 37% of the women were employed, and as employed women, they were more likely to travel outside their CA for VTP services. This supports the idea that financial resources and workplace constraints may influence healthcare-seeking behavior [36,37]. Marital status did not show a strong impact on mobility, with only slight differences observed between married/non-marital union women and others. Among women who decided to go outside their CA, 91% experienced a waiting time of more than 15 days between certification and the procedure, even in cases classified as more urgent (49% vs. 40%). Many of these women (75%) sought assistance from family counseling services, while 24% consulted a general practitioner. In contrast, among women who decided to stay within their CA, 30% preferred to consult a general practitioner. Although urgency and fetal malformations were more frequently observed among women who traveled outside their CA for VTP, these factors did not show statistical significance in the multivariate analysis. This suggests that while these conditions may contribute to mobility, they are not the primary driving factors, compared to socioeconomic and accessibility-related variables.

Our study has some limitations.

We acknowledge that our model does not account for the workload of hospitals related to conscientious objection, nor do we have access to data on waiting times for VTP procedures at each facility. These factors could significantly influence women’s decision-making, particularly if such information is known or perceived by patients. For example, awareness of longer waiting lists or limited availability due to staff objection might lead women to seek care in other hospitals. Information about service availability is typically provided during the first clinical visit, but we did not have access to these data. As a partial proxy, we considered the time interval between the certification, mandatory before accessing the procedure, and the actual intervention. However, we recognize that this measure does not fully capture the complexity of the scheduling process, as booking the procedure may depend on multiple organizational and individual factors. Although the expected waiting time should not exceed 15 days, variations may still occur and influence patient choices. The deprivation index used may not fully capture current socioeconomic conditions, as it is based on historical data. However, to mitigate this issue, we opted to classify deprivation levels using quartiles of the index distribution. This approach reduces the likelihood that socioeconomic changes over time would shift a municipality from one quartile to another, thus preserving the relative positioning of areas within the regional context. Additionally, missing data in administrative healthcare records could introduce potential biases. To address this, we conducted a missing data imputation analysis and verified that the presence of missing values did not significantly alter the estimates, thereby strengthening the robustness of our findings. Although regression analyses based on both the complete-case and imputed datasets were compared, the imputation did not yield any substantive improvements in the estimates. In most cases, confidence intervals widened and variance increased, reflecting the additional uncertainty introduced by the imputation process. The only notable exception was the estimate for the number of structures within the catchment area (CA), where imputation modestly improved precision. Consequently, considering the large sample size and with a low rate of missing data, our complete-case analysis appears robust and methodologically appropriate in this context [20,47,48]. Another limitation may be the inability to capture the full range of motivations influencing women’s decisions to seek VTP. The nature of the dataset does not allow for controlling or accounting for women’s perceptions of local healthcare facilities.

The Poisson regression model is widely used for modeling count data, such as the number of events occurring within a specified time period or spatial area. It is considered robust due to its capacity to provide reliable estimates even when certain model assumptions are not strictly met [38,39]. A major strength of our study is the large dataset available, providing a comprehensive view of VTP accessibility in the Apulia region. Moreover, our method for defining the CA, based on spatial and temporal distances, offers a more nuanced and realistic understanding of healthcare mobility patterns.

Future research should extend the study overtime to observe trends and changes in accessibility over the region or over the years. Additionally, further investigations should explore the role of a family counseling center, the impact of conscientious objection among healthcare providers, and the need for reducing stigma surrounding voluntary termination of pregnancy. These factors are essential for improving reproductive healthcare policies and ensuring equitable access to services.

## 5. Conclusions

The higher frequency of VTPs occurring outside the catchment area was found to be significantly associated with socioeconomic conditions, employment status ("employed"), educational level (diploma), and longer gestational age. In particular, medium and low levels of social deprivation and a lower number of hospitals with VTP service within the catchment area appear to be related to women’s mobility. These findings may support healthcare management in evaluating the distribution of VTP services, the types of procedures offered, and the perceived quality of services, with the aim of initiating a quality improvement process. Moreover, the results highlight the need for further research to gain a deeper understanding of the factors influencing women’s choice of facility. While this study relied on national survey data, a targeted questionnaire administered to a representative sample could provide more detailed insights. Such information could help refine access procedures and reduce disparities affecting women who are disadvantaged by their place of residence.

The availability of pharmacological abortion is not uniform across the Apulia region. Although this factor may influence women’s decision to access one facility over another for a medical abortion, it cannot be concluded that this is the sole or primary reason for seeking care outside their catchment area.

## Figures and Tables

**Figure 1 healthcare-13-02160-f001:**
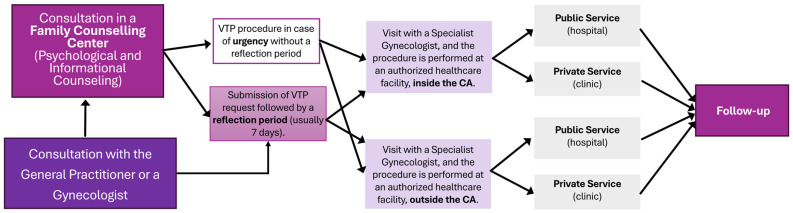
Structured pathway for women opting for voluntary termination of pregnancy in Italy.

**Figure 2 healthcare-13-02160-f002:**
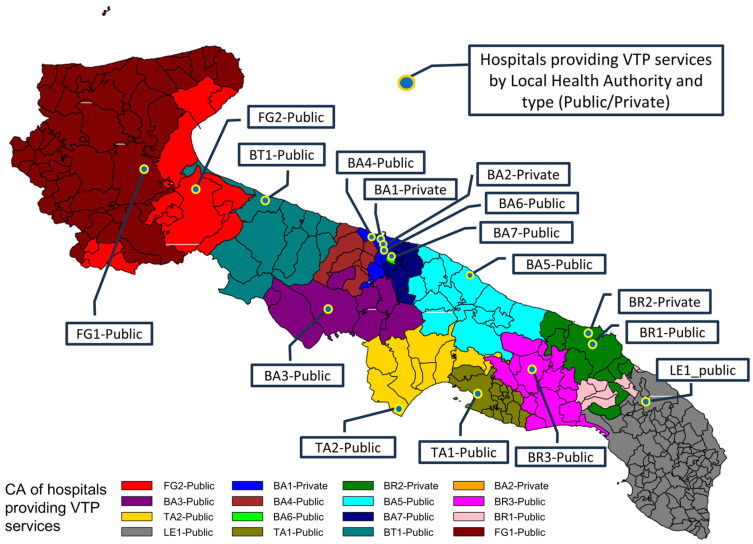
Geographic distribution of voluntary termination of pregnancy (VTP) hospitals and corresponding catchment areas (CAs) in Apulia.

**Figure 3 healthcare-13-02160-f003:**
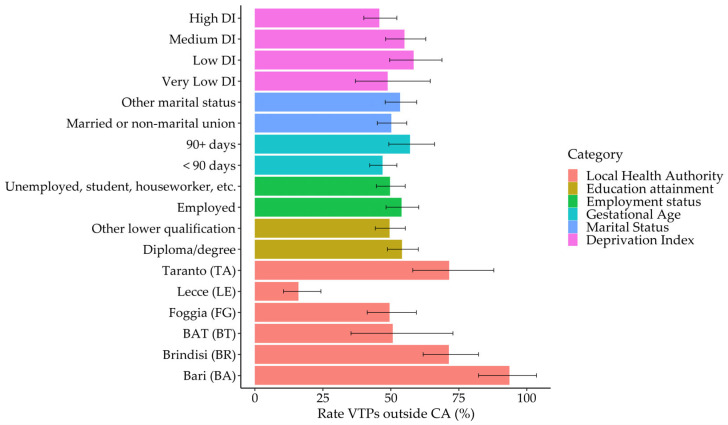
Estimated rates of undergoing VTP outside the catchment area (CA), with 95% confidence intervals, by sociodemographic and clinical characteristics. Estimates are derived from the multivariable Poisson model. Bars represent point estimates, and error lines indicate 95% confidence intervals.

**Figure 4 healthcare-13-02160-f004:**
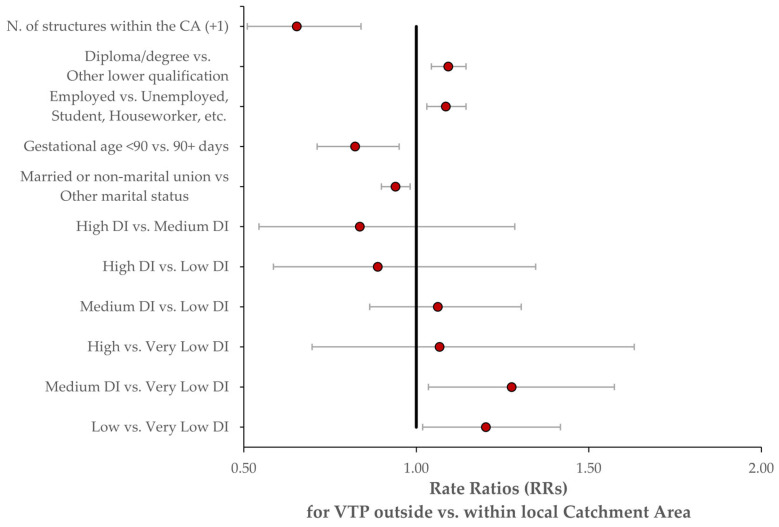
Forest plot of adjusted rate ratios (RRs) for mobility outside the catchment area, estimated from the multivariable Poisson model. Red dots represent point estimates, and horizontal bars represent 95% confidence intervals.

**Table 1 healthcare-13-02160-t001:** Sociodemographic characteristics of women seeking abortion services. Comparison between VTP practiced outside and within the catchment area.

Variable	Outside the CA n = 2657	Within the CA n = 2201	*p*
Age at pregnancy			
<30 years	1172 (44.1%)	972 (44.2%)	0.9711
≥30 years	1485 (55.9%)	1229 (55.8%)	
Educational attainment			
Diploma/degree	1558 (58.6%)	1129 (51.3%)	<0.0001
Other lower qualification	1099 (41.4%)	1072 (48.7%)	
Employment status			
Employed	1137 (42.8%)	676 (30.7%)	<0.0001
Unemployed, Student, Houseworker, etc.	1520 (57.2%)	1525 (69.3%)	
Citizenship			
Italian	2436 (91.7%)	1959 (89.0%)	0.0016
Other	221 (8.3%)	242 (11.0%)	
Marital Status			
Married or non-marital union	930 (35.0%)	807 (36.7%)	0.2286
Other	1727 (65.0%)	1394 (63.3%)	

Data are shown as frequency (percentages).

**Table 2 healthcare-13-02160-t002:** Organizational and administrative characteristics of healthcare facilities and clinical characteristics of VTP. Comparison between VTP practiced outside and within the catchment area.

Variable	Outside the CA n = 2657	Within the CA n = 2201	*p*
Type of hospital			
Private	491 (18.5%)	274 (12.4%)	<0.0001
Public	2166 (81.5%)	1927 (87.6%)	
The time interval between certification/authorization for the procedure and its completion			
>15 days	2481 (91.1%)	1880 (84%)	<0.0001
≤15 days	241 (8.9%)	358 (16%)	
The facility that releases the certification			
Family counseling center	2007 (75.5%)	1529 (69.5%)	<0.0001
General practitioner or other facility	650 (24.5%)	672 (30.5%)	
Type of VTP			
Pharmacological	1645 (61.9%)	945 (42.9%)	<0.0001
Surgical	1012 (38.1%)	1256 (57.1%)	
Urgency			
Yes	1317 (49.6%)	901 (40.9%)	<0.0001
No	1340 (50.4%)	1300 (59.1%)	
Gestational age			
<90 days	2454 (92.4%)	2109 (95.8%)	<0.0001
≥90 days	203 (7.6%)	92 (4.2%)	
Fetal malformations			
Yes	151 (5.7%)	74 (3.4%)	
No	2506 (94.3%)	2127 (96.6%)	

Data are shown as frequency (percentages).

## Data Availability

The data for the analysis were sourced directly from the Health Information System of the Apulia Region and are not publicly available for the nature of the information and due to privacy restrictions.

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
