# Peer review of "Geographical Distance, Socioeconomic Deprivation, and Educational Level Shape Access to Voluntary Termination of Pregnancy in a Southern Region of Italy [Author-notes fn1-healthcare-13-02160]"

_healthcare, 2025, doi:10.3390/healthcare13172160_

Round 1
Reviewer 1 Report (Previous Reviewer 3)
Comments and Suggestions for Authors
Dear Authors,
Thank you for the opportunity to review your manuscript. This is a strong and timely analysis and involves the innovative use of spatial-temporal measures to define catchment areas, with the work having clear policy relevance. The study design and statistical modelling are appropriate, and the linkage of sociodemographic characteristics to mobility outside catchment areas is clearly established. However, before publication, the manuscript would benefit from several important revisions.
The sensitivity analyses in Tables S1 and S2 are essential to demonstrating the robustness of your findings, but are not adequately reflected in the main text. Key results from the imputed data analysis in Table S2B differ in magnitude from the complete-case estimates. For example, the “Number of structures within CA” coefficient changes from 0.65 to 0.84, yet these differences are neither reported nor discussed. Given that the imputation analysis was performed specifically to test robustness, these numerical results and any meaningful differences should be presented in the Results section and acknowledged in the Discussion.
In interpreting the deprivation index findings, please clarify that the “High” deprivation category serves as the reference group. This is not clearly stated in the current text and may confuse the interpretation of the direction of the effects.
The terms “database without missing data” and “full database” should be replaced with precise descriptors such as “complete-case dataset” and “imputed dataset” to maintain clarity.
Figures and tables should be easier to interpret at a glance, with clear axis labels, descriptive legends, and consideration of more visual formats (e.g., forest plots) for regression results.
The Methods could be made more concise without losing essential detail; the rationale for complete-case analysis after imputation, for example, should be stated directly in one sentence before moving into results.
Overall, the analysis is sound, and the topic is of high importance. These revisions, alongside professional English editing, c ould improve the clarity, accessibility, and overall impact of the work.
Comments on the Quality of English LanguageThe manuscript is currently comprehensible but would benefit from language editing. Several sentences are overlong and contain awkward phrasing; articles are sometimes missing, and prepositions are occasionally misused. Professional editing can ensure grammatical accuracy, consistent verb tenses, and a polished academic tone please.
Author Response
Dear Authors,
Thank you for the opportunity to review your manuscript. This is a strong and timely analysis and involves the innovative use of spatial-temporal measures to define catchment areas, with the work having clear policy relevance. The study design and statistical modelling are appropriate, and the linkage of sociodemographic characteristics to mobility outside catchment areas is clearly established. However, before publication, the manuscript would benefit from several important revisions.
We sincerely thank the reviewer for the positive evaluation of our work and for recognizing the novelty and policy relevance of our approach. We also appreciate the constructive feedback provided. We have carefully considered the comments and revised the manuscript accordingly to further strengthen the clarity of our study. Below, we provide detailed responses to each of the reviewer’s points.
Comment 1: The sensitivity analyses in Tables S1 and S2 are essential to demonstrating the robustness of your findings, but are not adequately reflected in the main text. Key results from the imputed data analysis in Table S2B differ in magnitude from the complete-case estimates. For example, the “Number of structures within CA” coefficient changes from 0.65 to 0.84, yet these differences are neither reported nor discussed. Given that the imputation analysis was performed specifically to test robustness, these numerical results and any meaningful differences should be presented in the Results section and acknowledged in the Discussion.
Response 1: We truly value your advice. We thank the reviewer for this important observation. Following the suggestion, we have now explicitly reported the main findings from the sensitivity analyses in the Results section and expanded the Discussion to acknowledge and interpret the differences between complete-case and imputed analyses.
In the Results section, the following text:
“In 2022, in Apulia had been registered 5316 women that choose a VTP. We selected the 4997 (93.6%) women resided within the Apulia's municipalities and 4858 had no missing data in the record trace fields used for the analysis. The median of age of the women that sought a VTP was 31 years [IQR 24-36], and the median of weeks of amenorrhea was 7 [IQR 6-9].”
has been replaced with:
“In 2022, in Apulia 5,316 women chose a VTP. We selected the 4,997 (93.6%) women resided within the Apulia's municipalities. A total of 4,858 women were included in the complete-case analysis. The previous comparison between complete-case dataset and imputed dataset indicate that the decision criterion based on p‑values remains invariant across both datasets in the explorative analysis (Table S1A-B), and that the resulting estimates and conclusions are congruent (Table S2A-B). We observed no meaningful differences in the exp(β) estimates between the complete-case and imputed datasets. In the Poisson regression for the number of structures within the CA, confidence intervals (CI) and standard errors (SE) narrowed slightly (e.g., from [0.57-0.74] to [0.78-0.9] and from 0.07 to 0.03 respectively), while exp(β) increased from 0.45 to 0.84. Although some variation in point estimates was noted, but the directionality of all associations remained unchanged, and increases in estimates across the imputed dataset did not exceed 10%, confirming the robustness of the results in both cases.
Among the 4,858 women in the complete-case dataset, the median of age of the women that sought a VTP was 31 years [IQR: 24-36], and the median gestational age at the time of seeking a VTP was 7 weeks [IQR: 6-9].”
In the Discussion section, the paragraph:
“Additionally, missing data in administrative healthcare records could introduce potential biases. To address this, we conducted a missing data imputation analysis and verified that the presence of missing values did not significantly alter the estimates, thereby strengthening the robustness of our findings.”
was expanded as follows:
“Additionally, missing data in administrative healthcare records could introduce potential biases. To address this, we conducted a missing data imputation analysis and verified that the presence of missing values did not significantly alter the estimates, thereby strengthening the robustness of our findings. Although regression analyses based on both the complete-case and imputed datasets were compared, the imputation did not yield any substantive improvements in the estimates. In most cases, confidence intervals widened, and variance increased, reflecting the additional uncertainty introduced by the imputation process. The only notable exception was the estimate for the number of structures within the catchment area (CA), where imputation modestly improved precision. Consequently, considering the large sample size and with a low rate of missing data, our complete‑case analysis appears robust and methodologically appropriate in this context [20,46,47].”
Comment 2: In interpreting the deprivation index findings, please clarify that the “High” deprivation category serves as the reference group. This is not clearly stated in the current text and may confuse the interpretation of the direction of the effects.
Response 2: Your suggestion is highly appreciated. We have clarified the reference group for all variables, including the Deprivation Index (DI) category, below Figure 3 at line no. 412 by replacing it:
“Individuals with low and medium deprivation indices showed significantly higher RRs when compared to those with a very low deprivation index, with RRs of 1.20 (95% CI: 1.02 - 1.42) and 1.28 (95% CI: 1.03 - 1.57), respectively.”
with the following statement:
“In the regression models, we set the following reference categories: "High DI" for the deprivation index; single (or, more generally, unmarried) women for marital status; “Taranto (TA)” for the Local Health Authority (or province of residence); VTPs sought ≥ 90 days after presentation, for time until voluntary termination of pregnancy (VTP); unemployment, student status, etc., for employment status; and the lowest educational qualification as the baseline for education. Women with low and medium deprivation indices showed significantly higher RRs when compared to those with a very low deprivation index, with RRs of 1.20 (95% CI: 1.02 - 1.42) and 1.28 (95% CI: 1.03 - 1.57), respectively.”
Comment 3: The terms “database without missing data” and “full database” should be replaced with precise descriptors such as “complete-case dataset” and “imputed dataset” to maintain clarity.
Response 3: Thank you for the suggestion. We replaced all the terms with the ones indicated above.
Comment 4: Figures and tables should be easier to interpret at a glance, with clear axis labels, descriptive legends, and consideration of more visual formats (e.g., forest plots) for regression results.
Response 4: We thank the reviewer for this suggestion. In the revised version, we have improved the readability of all figures and tables by ensuring clearer axis labels and more descriptive titles. In particular, Figure 4 already presents the regression results in a forest plot format, as recommended. We have revised the figure to further enhance its clarity, making the axis labels more explicit and modifying the title.
Comment 5: The Methods could be made more concise without losing essential detail; the rationale for complete-case analysis after imputation, for example, should be stated directly in one sentence before moving into results.
Response 5: We appreciate your valuable feedback. We present the rationale for adopting complete‑case analysis and concisely emphasize key methodological details, replacing the following paragraph:
“VTP cases with missing data in at least one of the analyzed variables were excluded from the analysis. Before excluding these cases, we ensured that the missing values in the VTPs data flow were randomly distributed. This was performed through a missing imputation procedure using the FCS (fully conditional specification) method for categorical variables and by comparing the analysis results between the database excluding missing values and the database after imputation. Specifically, the relationships between the outcome (inside/outside catchment area) and each variable were compared to verify the consistency of the results, and then the causality of the missing data was examined. Sensitivity analyses using the Fully Conditional Specification (FCS) method for missing data imputation confirmed the robustness of the findings (see Supplementary Tables S1 and S2). Based on these results, the final models were estimated using the complete-case dataset. Additionally, a missingness pattern matrix is provided in the Supplementary Material (Table S3) to enhance transparency regarding the extent and structure of missing data.”
with this new paragraph:
“VTP cases with missing data in any analyzed covariates were excluded, although the missingness mechanism could not be definitively determined for each individual variable. This assessment involved comparing results from a complete-case dataset and an imputed dataset—using Fully Conditional Specification (FCS) for categorical variables—and conducting sensitivity analyses that affirmed the robustness of results across both datasets (see Supplementary Tables S1–S2). Given a large sample size (only 3% of incomplete cases), we therefore based final model estimation on the complete-case dataset to avoid potential imputation induced noise and preserve estimate precision [20]. Additionally, a missingness pattern matrix is provided in the Supplementary Material (Table S3) to enhance transparency regarding the extent and structure of missing data.”
Overall, the analysis is sound, and the topic is of high importance. These revisions, alongside professional English editing, could improve the clarity, accessibility, and overall impact of the work.
Answer to the Reviewer: Thank you for your thoughtful recommendations, we will send the paper for English Language editing.
Reviewer 2 Report (Previous Reviewer 2)
Comments and Suggestions for Authors
Thank you for revising the manuscript. It is vastly improved.
Author Response
Thank you for revising the manuscript. It is vastly improved.
We thank the reviewer because his comments allowed us to significantly improve the manuscript.
This manuscript is a resubmission of an earlier submission. The following is a list of the peer review reports and author responses from that submission.
Round 1
Reviewer 1 Report
Comments and Suggestions for Authors
This manuscript presents a retrospective observational study evaluating the influence of geographical distance, socioeconomic deprivation, and individual characteristics (education, employment) on women's mobility in accessing voluntary termination of pregnancy (VTP) services in the Apulia region of Italy. The study defines healthcare catchment areas (CAs) using spatial and temporal travel metrics.
- The current title is clear, concise, and accurate; however, a minor revision to add geographic specificity would enhance clarity and searchability without altering the meaning.
- The abstract mentions findings (e.g., mobility was more frequent) without providing numerical effect sizes or p-values, which would strengthen scientific transparency. Suggest adding one or two rate ratios or confidence intervals to support major claims. The term “women’s choices tend to converge towards healthcare facilities in their catchment area” could be confusing since the results emphasize mobility away from the CA. Suggest clarifying the direction of analysis more precisely: e.g., to evaluate factors influencing women’s decisions to seek VTP services outside their local catchment area. Geographic context (Apulia, Italy) is not mentioned in the abstract. Including the region would improve contextual relevance for readers.
- In the introduction better to include citing more region-specific studies if available.
- This design is appropriate to investigate associations between geographic, socioeconomic, and individual factors and mobility for accessing VTP. However, the method for assigning a reference facility assumes the closest spatial-temporal option is preferred. This assumption may not account for patient preferences (e.g., quality, stigma, abortion method). The temporal distance was calculated via Google Maps, but traffic variability, public transport access, or car ownership could affect true accessibility. A sensitivity analysis or stratification by urban/rural settings would strengthen the methodology. The term "Poisson regression for repeated measures" is used, but it's not entirely clear which repeated measurements are considered (e.g., multiple patients per municipality or multiple facilities?). A brief justification for why age was dichotomized at 30 years would improve transparency.
- In analysis, while results are clear, a supplemental table showing full model coefficients would strengthen transparency for readers seeking in-depth analysis.
- Discussion is well-integrated with existing literature, but briefly outlines actionable steps like expanding access to pharmacological abortion in underserved areas or targeting interventions in high-deprivation municipalities. Consider integrating a short paragraph connecting this structural barrier more directly to mobility and patient experience. Some portions of the discussion repeat information from the Results or Introduction (e.g., the structure of the Italian abortion pathway or definitions of CAs).
- The conclusion could be strengthened by briefly stating why these findings matter, e.g., their implications for health service planning, equity in reproductive rights, or addressing access barriers in underserved areas.
- There is an inconsistency in citation style and outdated references. online sources (e.g., [1], [2], [26]), could be improved. Reference [3], a 2017 article on “Social awareness on Voluntary Termination of Pregnancy,” is relevant but might not be as strong methodologically as others. Reference [39] refers to a conference proceeding version of the current article. It should be clearly labelled to avoid self-plagiarism concerns.
The English language is acceptable for publication but would benefit from professional copyediting to improve fluency, correct minor grammatical issues, and enhance overall readability.
Reviewer 2 Report
Comments and Suggestions for Authors
Thank you for the opportunity to review this study. It is an interesting analysis of pregnant women's choices for a healthcare facility. The manuscript will benefit from revisions to certain portions of the methods, results and the discussion section.
While the authors provide a good background of the logistics of a VTP, perhaps they can provide some figures on how many women in Italy undergo this procedure in a year and whether this figure has changed over the years.
The authors need to explain this statement further - "For many women, undergoing an abortion is emotionally challenging prompting them to seek care outside their residential area if access is perceived as difficult or less reliable". In the Discussion section they also mention that "conscientious objection and regional disparities in service availability may influence patient decisions". However, analytical models do not control for women's perceptions of local facilities and the authors do not discuss the this in the limitations section.
Relatedly, this statement in the Discussion section contradicts the main premise of the study - "This variability among LHAs may be partly explained by the distribution and density of available VTP facilities: in areas such as Bari, where multiple facilities are located within close proximity, women may choose to access services outside their assigned CA while still remaining within a short geographical distance." If this is the case, then what is the point of assigning CAs to facilities? Why not just use the metric of 'distance to nearest facility'? Perhaps the authors meant to highlight some other point here. Please confirm whether that is true. Also, the authors should include in an Appendix a map of Apulia that shows the location of all facilities, the centroids of each municipality and the exact CA of each facility.
Please specify the exact source of the data. Was it electronic medical record systems or insurance claims or something else? The authors mention in the discussion that the data are from administrative healthcare records, has it been used for similar research before?
Results from sensitivity analyses under the FCS method should be included in an Appendix.
The method of assigning CAs seems intuitive but the authors need to provide other studies that have used it or something similar.
If possible - an alternate study outcome could be 'distance to the VTP facility from residence' rather than a binary outcome of 'inside or outside the CA'.
In Table 1 - what is the Other employment status exactly? Is it supposed to be 'unemployed'?
Full results from the Poisson model should be included in an Appendix. Also, why was pairwise comparison done after running multivariate regression models?
Finally, a major determinant of where a woman chooses to receive VTP is availability. If the nearest facility does not have the capacity to perform VTP, then a woman will be forced to go elsewhere. This should be mentioned as a limitation.
Reviewer 3 Report
Comments and Suggestions for Authors
Thank you for this interesting and valuable manuscript. The subject matter is both timely and important. Understanding the determinants of mobility for abortion care, especially in light of ongoing health service rationalisation and regional disparities, is critical to informing equitable reproductive health policy in Italy and around the world. The study's use of modelling and administrative health data represents a valuable contribution.
However, I must express serious concerns, as there is extensive overlap of text and concepts with previously published material, specifically the Springer volume(DOI: https://link.springer.com/book/10.1007/978-3-031-64350-7). Large sections of your introduction and methods appear to be directly lifted or only lightly paraphrased from this source without adequate attribution or quotation to the source you have taken this from, as while you cite, these are the original chapter's citations, not your own. Significant sections have only a few words changed here and there, and this constitutes a serious academic integrity violation. The plagiarism includes not only factual descriptions but also structure and phrasing, accounting for approximately one-fifth of the manuscript. Any future version of this work must address this by thoroughly rewriting affected passages in your own words and providing appropriate citations where concepts or structure are derived from previous literature.
Next, while the use of Poisson regression to examine the determinants of healthcare mobility is broadly appropriate, the methodological rationale would benefit from further elaboration. Specifically, the decision to employ Poisson regression is inadequately justified in the context of alternative models that may also be suitable, such as negative binomial regression if overdispersion were present. The manuscript should clarify how the model's assumptions were verified and discuss why repeated measures were used, as well as how clustering was handled, since these have direct implications for variance estimation and model validity. More detail is needed, generally.
Moreover, the strategy for covariate selection, moving from univariate to multivariate analysis, is standard but would benefit from a more precise explanation of whether model selection criteria or checks for multicollinearity were conducted. Given the correlation between education, employment, and deprivation indices, it is crucial to confirm that multicollinearity did not bias the estimated associations. Additionally, the dichotomisation of continuous variables (e.g. age and education), although common, raises concerns about the loss of information and potential arbitrary threshold effects. Justifying these thresholds theoretically or empirically would enhance interpretability and rigour.
Regarding missing data, although the manuscript mentions using FCS for imputation and comparing imputed versus complete-case results, it remains unclear whether the final models were run on the imputed dataset or only on cases with complete data. Providing more transparency about the imputation diagnostics, missingness patterns, and justification for including or excluding imputed data is essential to assess the robustness of the findings. Again, it is about providing more detail.
A more apparent distinction between descriptive findings and model-based inferences in the results and discussion would enhance the study's interpretability and help prevent readers from overinterpreting the raw group differences. While descriptive statistics are valuable for providing context, their juxtaposition with adjusted regression results without sufficient differentiation could lead some readers to infer causality (rather than just associations). Additionally, concerning catchment areas, which combine spatial and temporal proximity, this notable methodological strength involves relying on municipal centroids to estimate distances. However, this approach may not fully account for variations in accessibility within larger or rural municipalities, where intra-municipal travel times can differ significantly. Acknowledging this limitation would further reinforce the robustness of the spatial analysis. Again, this is about added detail and richer explanation.
Finally, although the manuscript rightly draws attention to inequalities in service availability and conscientious objection, these structural issues could be addressed in greater detail, particularly in the implications and recommendations. How might health planners interpret your findings? What reforms could reduce inequity in abortion access, and how might your model be applied to other regions of Italy, Europe, or elsewhere?
Overall, there is a good foundation here, but numerous issues, and this article cannot be published with so much content directly taken from another source in the way it has been. This must be reworded into your own words, and your own literature search and referencing undertaken.
Comments on the Quality of English LanguageI encourage improvements in the clarity and fluency of the manuscript's expression. At times, sentence structure is overly long and indirect, and several key terms are inconsistently defined or used interchangeably (for instance, “outside the catchment area” and “mobility” are not always clearly distinguished). The abstract and conclusion would also benefit from more precise language to accurately reflect the findings and their limitations, thereby improving clarity.